# ATP and NAD^+^ Deficiency in Parkinson’s Disease

**DOI:** 10.3390/nu15040943

**Published:** 2023-02-14

**Authors:** Laurie K. Mischley, Eric Shankland, Sophia Z. Liu, Saakshi Bhayana, Devon J. Fox, David J. Marcinek

**Affiliations:** 1Translational Bioenergetics Laboratory, Department of Radiology, University of Washington, Seattle, WA 98105, USA; 2Parkinson Center for Pragmatic Research, Seattle, WA 98133, USA; 3Department of Laboratory Medicine and Pathology, University of Washington, Seattle, WA 98195, USA

**Keywords:** neurodegenerative, bioenergetics, magnetic resonance spectroscopy, deficiency, metabolic perturbation, mitochondrial respiratory chain, complex I, NAD

## Abstract

The goal of this study is to identify a signature of bioenergetic and functional markers in the muscles of individuals with Parkinson’s disease (PD). Quantitative physiological properties of in vivo hand muscle (FDI, first dorsal interosseus) and leg muscle (TA, Tibialis Anterior) of older individuals with PD were compared to historical age/gender-matched controls (N = 30). Magnetic resonance spectroscopy and imaging (MRS) were used to assess in vivo mitochondrial and cell energetic dysfunction, including maximum mitochondrial ATP production (ATPmax), NAD concentrations linked to energy/stress pathways, and muscle size. Muscle function was measured via a single muscle fatigue test. TA ATPmax and NAD levels were significantly lower in the PD cohort compared to controls (ATPmax: 0.66 mM/s ± 0.03 vs. 0.76 ± 0.02; NAD: 0.75 mM ± 0.05 vs. 0.91 ± 0.04). Muscle endurance and specific force were also lower in both hand and leg muscles in the PD subjects. Exploratory analyses of mitochondrial markers and individual symptoms suggested that higher ATPmax was associated with a greater sense of motivation and engagement and less REM sleep behavior disorder (RBD). ATPmax was not associated with clinical severity or individual symptom(s), years since diagnosis, or quality of life. Results from this pilot study contribute to a growing body of evidence that PD is not a brain disease, but a systemic metabolic syndrome with disrupted cellular energetics and function in peripheral tissues. The significant impairment of both mitochondrial ATP production and resting metabolite levels in the TA muscles of the PD patients suggests that skeletal muscle mitochondrial function may be an important tool for mechanistic understanding and clinical application in PD patients. This study looked at individuals with mid-stage PD; future research should evaluate whether the observed metabolic perturbations in muscle dysfunction occur in the early stages of the disease and whether they have value as theragnostic biomarkers.

## 1. Introduction

Parkinson’s disease (PD) is the fastest growing neurological disorder globally, and the burden is predicted to double from six million to over 12 million between the years 2015 and 2040 [1]. First described more than two centuries ago, there are still no established therapies to slow, stop, or reverse the disease; the mainstay of medical management is to use dopaminergic therapies to temporarily reduce motor symptoms of the disease [2]. A variety of genetic and environmental variables contribute to PD pathophysiology, including oxidative stress, protein aggregation, inflammation, impaired autophagy, and mitochondrial dysfunction, and each of these has been proposed as a possible therapeutic target [3].

Mitochondrial dysfunction was first demonstrated in individuals with PD in the late 1980s and has since been demonstrated in both sporadic and familial forms of PD [4,5]. Mitochondria generate ATP to meet the majority of the cellular energy demands via oxidative phosphorylation, which uses metabolites generated by the tricarboxylic acid cycle (TCA) driving ATP production through the electron transport system (ETS). The ETS produces the proton gradient across the inner mitochondrial membrane used to generate ATP [6]. Mitochondria play an important role in regulating cellular stress and adaptive signaling through modulation of cell redox and metabolite homeostasis [7]. Mitochondrial complex I deficiency in PD has been well-described for over three decades [4,8]. An age-related decline was observed in complexes I and IV activity from postmortem brains of individuals with PD and Alzheimer’s disease (AD), with less robust evidence supporting deficits in major depressive disorder, bipolar disorder, and schizophrenia [9]. Defects in complex I of the ETS appear to be especially associated with PD and PD-like symptoms in animal models. For example, rotenone, a complex I inhibitor, is used to create rodent models of PD in the lab. Rotenone causes the accumulation of Lewy bodies in the nigrostriatal system and clinical features of parkinsonism [10]. Mitochondrial dysfunction, especially at complex I, is now understood to be central to PD pathophysiology and improvement in cellular respiration has the potential to prevent neuronal loss and improve clinical outcomes [5].

Magnetic resonance spectroscopy (MRS) is a noninvasive method of measuring cellular energy metabolism in vivo in a range of human tissues, including brain and muscle. MRS uses phosphorus (^31^P) to determine mitochondrial ATP production as well as basal metabolic states in skeletal muscle [11,12,13,14]. Using ^31^P MRS, putamen and midbrain ATP have been shown to be decreased in individuals across PD stages, but less is known about skeletal muscle mitochondrial function in PD [15]. A 1994 study of finger flexor muscles of seven patients with PD was unable to demonstrate a difference in cellular energetics when compared to eleven healthy controls [16]. In skeletal muscle, the rate of recovery of phosphocreatine (PCr) can be used to calculate the maximum rate of mitochondrial ATP production (ATPmax). In human skeletal muscle, ATPmax is correlated with mitochondrial content and maximum in vitro respiration and thus provides a non-invasive measure of mitochondrial capacity [17,18]. This approach has been used to demonstrate declines in skeletal muscle ATPmax in aging skeletal muscle and in other chronic disease conditions [18,19,20].

Nicotinamide adenine dinucleotide (NAD) is a coenzyme consisting of two nucleotides joined through a diphosphate bridge; it exists in two forms, oxidized (NAD) and reduced (NADH). NAD is an oxidizing agent, able to accept electrons from other molecules in the cell. NAD can be synthesized endogenously from dietary niacin and recycled as part of the salvage pathway. The NAD^+^/NADH ratio is essential to maintaining redox balance of the cell, and reduced concentrations of NAD are linked to poor metabolic health in aging and chronic diseases [21,22]. Elevating NAD levels by targeted supplementation has improved health and reversed pathology in animal models of disease [23,24], and NAD enhancing supplements are currently being tested in clinical trials for multiple human conditions [25,26,27]. The therapeutic use of NAD+ precursors, such as nicotinamide riboside (NR), as a therapeutic target to compensate for deficits in complex I, is particularly appealing in PD [28]. NR was shown to prevent neuronal cell loss in patient-derived cell lines and Drosophila models of GBA-related PD [29]. NADPARK, a hallmark translational study, recently showed viability for NR as a potential therapeutic strategy in humans with PD [30].

The goal of this pilot study was to test whether metabolic and functional deficits in skeletal muscle tissue could be measured in individuals with PD using ^31^P MRS. ATPmax and muscle function were measured in the hand and leg muscles, and NAD metabolites were measured in the leg muscles of 30 PD subjects and compared to a healthy control dataset collected at the Translational Bioenergetics Laboratory [TBL] at the University of Washington. Exploratory analyses were performed to evaluate whether mitochondrial dysfunction was related to PD symptomatology or quality of life.

## 2. Materials and Methods

### 2.1. Participants and Study Design

Thirty individuals with PD between 65 and 85 years old were recruited for a single study visit. All study procedures took place at the Translational Bioenergetics Laboratory (TBL) human research lab in the UW Department of Radiology in Seattle, WA, USA. Control data from healthy age-matched subjects were collected between 2014–2021 during three previous clinical trials related to MRS muscle metabolomics at the UW Department of Radiology [31,32,33].

Direct data entry was utilized where possible with a REDCap database, hosted by the secure server at the University of Washington Institute for Translational Sciences (ITHS) [34,35]. REDCap (Research Electronic Data Capture) is a secure, web-based software platform designed to support data capture for research studies, providing (1) an intuitive interface for validated data capture; (2) audit trails for tracking data manipulation and export procedures; (3) automated export procedures for seamless data downloads to common statistical packages; and (4) procedures for data integration and interoperability with external sources.

Participants were instructed to take medications as they normally would on the study day. Participants were asked to avoid strenuous exercise one day before and prior to the study visit and asked to avoid large meals for at least one hour prior to the study visit. Dopamine equivalents were calculated using the following resource accessed at the study visit, between 28 September 2020 and 9 August 2021: https://www.parkinsonsmeasurement.org/toolBox/levodopaEquivalentDose.htm.

All rstudy participants were 65–85 years old, to match the age range of the control subjects (62–89 years old), had a Hoehn & Yahr Stage 2–3 (bilateral disease, not severely disabled), and claimed to be able to keep their right arm and right leg still (no tremor or dyskinesia) for approximately 90 min. PD diagnosis was supported by the presence of at least 3 of the following: symptoms that originally started on one side; tremor at rest (in any body part); progressive worsening of symptoms over time; levodopa improving symptoms by 70–100%; levodopa-response for five or more years; clinical course of ten years or more. Participants were ineligible if they had other serious health conditions or a diagnosis of dementia. Participants denied a history of epilepsy, stroke, brain surgery, or structural brain disease, any other serious illnesses, current drug or alcohol use or dependence, or cognitive impairment. Individuals enrolled in a clinical trial involving an investigational product or device were excluded. A 30-day washout was required for anyone supplementing with NAD, nicotinamide mononucleotide (NMN), nicotinamide riboside (NR), and other nutraceuticals designed to target NAD. All participants denied any acute infection in the 30 days prior to the study visit. Individuals were excluded if they had any contra-indication to magnetic resonance imaging, including pacemaker, pacemaker wires, aneurysm clip, or any electronic implant, weight over 136 kg (300 lb), metal embedded in soft tissue or in the eye, prosthetic eye, claustrophobia, substance abuse, use of recreational drugs, pregnancy, or other medical contraindications.

This study was approved by the University of Washington (UW) Institutional Review Board (IRB # STUDY00007024). Written and informed consent was obtained from all participants by study investigators. This study followed Good Clinical Practice guidelines. The trial is registered at Clinicaltrials.gov, identifier: NCT04300608.

Recruitment occurred through the Michael J Fox Foundation Trial Finder, the Washington PD Registry, internet/social media, regional support groups, and local provider offices. Efforts were made to enroll participants across genders, ethnicities, and geographic regions throughout the region who were reasonably representative of the population. To serve this pragmatic goal, participants were permitted to be using pharmaceutical and nutritional supplements, and be in other studies, except for those designed to boost NAD, as listed in the exclusion criteria. Participants were screened on the phone and those likely to qualify were invited to the UW to be physically evaluated. Ninety-seven individuals were screened on the phone, 33 of whom were invited to the UW to provide 30 individuals with PD capable of completing the study. Reasons for exclusion upon arrival included: Hoehn & Yahr 4 (n = 1), uncontrollable hand tremor (n = 1), or lacking the physical strength and agility to get in and out of the MRI machine (n = 1). A STROBE flow diagram for the study is provided in Figure 1.

Muscle endurance (i.e., resistance to fatigue) was determined on the right hand and leg skeletal muscles using custom-built exercise apparatus [33,36]. Muscle fatigue resistance was tested by measuring the number of contractions by the hand (*first dorsal interosseous*, FDI) and leg (*tibialis anterior*, TA) during repeated isometric contractions until exhaustion. The contraction intensity was set at 70% of the maximum voluntary contraction (MVC) and the exercise began at a frequency of 60 contractions per minute (bpm) for the first minute. This frequency increased at a rate of 10 bpm with each minute until exhaustion. Muscle strength defined as the MVC was measured as the average of 3 maximum contractions separated by 5 s with each one sustained for 3–5 s. Muscle specific force was determined by dividing maximal voluntary contraction force by muscle cross sectional area measured by magnetic resonance imaging (details below) (i.e., MVC/CSA).

### 2.2. Magnetic Resonance Spectroscopy

All NMR data were acquired in a 20 cm diameter, short bore [1 m], horizontal 4.7 Tesla magnet [Bruker BioSpin GmbH, Ettlingen, Germany]. Acquisition was accomplished via a Bruker AVIII console. All data acquisition utilized a dual-tuned, single-loop, ^31^P/^1^H surface coil [oval, major/minor axes: 7 cm/4 cm for the TA and 3 cm/2 cm for the FDI]. Field homogeneity was adjusted via gradient field mapping, providing high order [to 3rd order + Z^4^] shim correction. Following application of map result, x, y, and z shims were iteratively adjusted using increase in ^1^H fid [50 ms pulse (~45°), 30 kHz sweep, 2 s recycle] sum for feedback directing manual shim adjustment. The average observed line width for ^31^PCr was approximately 13 Hz [sd = 3.2]. Fully relaxed, decoupled [GARP], ^31^P spectra were acquired following a 50 msec, nominal 90° square pulse with spectral width 10,000 Hz into 4096 complex points. Thirty-two such pulse/acquire averages were repeated with a 10 s recycle time, during which no decoupler was applied to minimize differences in nOe. Time resolved data, acquired for exercise induced metabolite changes, were acquired as relaxed data with the following changes: dynamic spectra, 128 spectra/exercise bout, were acquired as a 2d spectrum [4096 × 128] with a 1.5 s [approximate Ernst angle] recycle time for 4 averages/spectrum [6 s].

### 2.3. NMR Data Processing

*PR recovery kinetics*: Each 2-dimensional exercise data set was imported into MestReNova and broadened [20 Hz for FDI, 10 Hz for TA]. Timecourse behavior of PCr and Pi was extracted via a script designed to integrate a fixed chemical shift subregion per resonance. Text output from these integrals was subsequently imported to an Excel spreadsheet with macros to calculate non-linear, least squares, exponential fits for PCr recovery [37]. Pi behavior was separately assessed for quality control.

*Nicotinamide quantitation by NMR spectroscopy*: Each raw, relaxed fid, imported into MestReNova [Mestrelab Research, SL, Santiago de Compostela, Spain], was subject to 5 Hz exponential broadening enhancing signal/noise and baseline corrected with a Whittaker smoother removing bone signal prior to exporting the region −9–−12 ppm [PCr = −2.54 ppm] to text for subsequent analysis with the OriginPro [OriginLab Corp, Northhampton, MA, USA] based model.

NAD was fit within the modified OriginPro framework [23], which produced optimized/integrated ^31^P spectra for the region −9–−12 ppm using the components alpha—ATP, NAD^+^, and NADH. The chemical shifts of the system were held relatively constant, having been set by spectral features from phantom samples at physiologic pH/[Mg^2+^]. As a group, the peak positions were allowed to vary up to +/− 0.1 ppm to allow the model to optimize a residual. All the components were modelled as Lorentzian with a single optimized linewidth. This simple model mimics other attempts to quantitate pyridine nucleotides in, e.g., brain ^31^P spectroscopy [38,39].

As previously described [23], alphaATP was fit as a doublet at −9.96 ppm with the coupling constant fixed at 16.8 Hz, based upon previously obtained, high resolution, in vivo, TA data from control subjects [unpublished results]. NAD^+^ was fit as a doublet of doublets with chemical shifts of −10.57 and −10.87 ppm with 20.03 Hz coupling. NADH was fit as a singlet with a chemical shift of −10.6 ppm.

### 2.4. Muscle Size by Magnetic Resonance Imaging

FDI and TA muscle cross-sectional areas (CSA) were determined from magnetic resonance images acquired as axial plane, T1-weighted, 2-D, gradient-echo images collected with the following parameters: 500 ms repetition time, 2.5 ms echo time, 3 mm slice thickness 1 mm inter-slice interval, 192 × 192 matrix, and number of excitations = 2. Five slices of each right limb were analyzed with NIH Image software (ImageJ, version 1.50 e) using manual planimetry to determine the muscle CSA.

### 2.5. ATPmax

The mitochondrial oxidative phosphorylation capacity (ATPmax) was determined as described [31]. Briefly, a short exercise bout (20–30 s) involving the index finger for hand (FDI muscle) pushing against a bar and dorsiflexion of the foot (TA muscle) was used to reduce PCr 30~50% from the resting state while maintaining muscle pH > 6.8. The PCr recovery was measured over 6 min to determine a time constant of recovery (t_PCr_) to yield ATP_max_ (= PCr_rest_/t_PCr_) where PCr_rest_ = 25.5 mM).

### 2.6. Statistical Analysis

A sample size of 30 individuals was calculated to achieve statistical significance (*p* < 0.05) for a 30% difference between groups, given the +/−10% error in our measurements. For this pilot study, no adjustments were made for multiple comparisons. Unpaired *t*-tests with correction for unequal variances were used for comparing NAD, ATPmax bioenergetic, and muscle function parameters between the control and PD groups. Sample sizes for the control groups vary for the physiological measurements because not all data were collected for every study or because of the exclusion of data that failed quality control tests as described in the original study [PMC6204600; PMC8282018; PMC8777576]. Sample sizes for the NAD quantitation are more limited due to the more stringent data quality limits necessary to reliably quantify NAD levels from the ^31^P MRS data. To ensure reliable quantitation of the NAD peaks, we limited our analyses to spectra with a Lorentzian linewidth (full width at half maximum amplitude) for the PCr peak. Deviation from Lorentzian shape and broader linewidths does not affect PCr recovery rate used to calculate ATPmax because this determination relies on changes in the relative peak area.

R was used to conduct the analysis of the correlation between PD symptoms and bioenergetic parameters. A Shapiro-Wilks test was used to determine normality. Square root transformations were completed on non-normally distributed measures. Some Simple linear regressions were conducted for predictors of Leg [NAD+], Leg ATPmax, and Hand ATPmax, outcome measures of UPDRS (parts 1–4 and total), PRO-PD total, and individual items of slowness, constipation, walking, freezing, falling, rising, daily living, motivation, depression, interest, anxiety, fatigue, daytime sleepiness, dyskinesia, tremor, balance, temperature regulation, orthostasis, visual disturbances, insomnia, REM sleep disturbances, dystonia, speech impairment, drooling, stooping posture, memory, comprehension, sense of smell, sexual dysfunction, medication side effects, urinary symptoms, hallucinations, and nausea, and PROMIS quality of life scale individual and total items.

## 3. Results

Of the 30 individuals enrolled in the MRI portion of the study, five measurements were excluded for hand imaging and two for leg imaging due to involuntary movements associated with PD. The distribution of participant characteristics can be found in Table 1.

### 3.1. Mitochondrial ATP Production

To test whether PD was associated with mitochondrial dysfunction in the skeletal muscle the ATPmax was assessed in the FDI and TA muscles in the PD subjects and compared with data from healthy age-matched controls from previously published clinical trials [31,32,33]. In the TA, the ATPmax was significantly reduced in the PD subjects relative to the control group (Figure 2A, Table 2), while there was no difference in ATPmax in the FDI between the PD and control subject groups (Figure 2B, Table 2). Since Parkinson’s disease symptoms frequently present asymmetrically, especially at the stage assessed in this study, we performed a secondary analysis to test whether the predominantly affected side was more impacted than the less affected side of the body. An analysis was done on the subgroup who reported their symptoms starting on the right vs. left (all our evaluations were done on the right) (Appendix A). There was no difference in the ATPmax values when compared in this way, suggesting that the decline in ATPmax was not due to a change in muscle activity based on PD symptoms.

### 3.2. Nicotinamide Metabolites

To assess whether PD was associated with a disruption of resting metabolic state, we attempted to quantify the NAD^+^, NADH, and NADt [total NAD] levels in resting muscle from the fully-relaxed ^31^P MRS data. Nicotinamide metabolites were only analyzed in the TA muscle because the lower signal quality in the hand prevented a reliable quantitation of NAD in this muscle. The mean NAD^+^ was significantly reduced in the PD vs. the control subjects (Figure 3 and Table 2) [*p* = 0.032]. Spectral modeling of this PD data clearly finds evidence of NADH. A series of phantom solutions were analyzed to establish detection limits [see Appendix A]. This analysis resulted in a limit of detection [LOQ] of 0.26 mM. The majority of the values were below the limit of quantitation; therefore, we excluded NADH from further analyses.

### 3.3. Muscle Function

To test whether the metabolic defects in PD patients were associated with skeletal muscle dysfunction, we measured maximum muscle force and muscle endurance with voluntary contractions in the hand and leg. Muscle specific force was determined by dividing the maximum force by the cross-sectional area of the muscle determined from MR images (Table 2). Maximum specific force was decreased in both the TA and FDI in PD patients compared to controls (Figure 4A,C). Like maximum force, muscle fatigue resistance was decreased in PD subjects in both muscles (Figure 4B,D). As for ATPmax we performed a secondary analysis to test whether the predominantly affected side was more impacted than the less impacted side in terms of muscle function. In agreement with the ATPmax data, there was no effect of right or left side in muscle fatigue resistance (Appendix A).

### 3.4. Relationship to PD Symptoms

Simple linear regressions were used to test if NAD, Leg ATPmax and Hand ATPmax significantly predicted UPDRS (parts 1–4 and total), PRO-PD, including individual items of the PRO-PD, and PROMIS individual and total items. NAD in the leg and ATPmax in both hand and leg were significantly inversely correlated to PD symptom severity when individual items were assessed.

Leg NAD, after transformation, was significantly negatively correlated with interest (*p* = 0.04), and REM sleep disturbances (*p* = 0.04). Motivation was negatively correlated but non-significantly (*p* = 0.07). These data suggest that greater NAD+ is associated with reduced symptom burden for interest and REM sleep disturbances and may be associated with greater motivation. Leg ATPmax, after transformations were significantly correlated with PRO-PD motor sub-score (*p* = 0.02), slowness (*p* = 0.05), constipation (*p* = 0.02), walking (*p* = 0.03), falling (*p* = 0.01), tremor (*p* = 0.03), balance (*p* = 0.03), visual disturbances (*p* = 0.03), and nausea (*p* = 0.01). These data suggest greater capacity for mitochondrial ATP production may reduce symptom severities of constipation, freezing, falling, nausea, tremor, balance, and stooping. Hand ATPmax after transformations were significantly correlated with orthostasis (*p* = 0.04) and the UPDRS part three item of rigidity (*p* = 0.03). No other symptoms were seen to have significant correlation (Appendix A).

## 4. Discussion

Mitochondrial defects associated with parkinsonism can be the result of genetic variants or environmental and metabolic insults [5,40]. Despite interest in the role of mitochondrial dysfunction in the etiology of PD, few studies have examined the presence of mitochondrial defects in peripheral tissues. We take advantage of the ability to use ^31^P MRS to non-invasively interrogate mitochondrial function in skeletal muscle to test whether PD is associated with mitochondrial dysfunction and metabolic disruption in peripheral tissues. We identified significant mitochondrial dysfunction in the TA muscle compared to a historical dataset from healthy age-matched controls. Deeper analysis of the resting MRS spectra also indicated disruption of NAD metabolism in the PD subjects. These metabolic defects were associated with poor skeletal muscle function in both the hand and leg muscles, suggesting these cellular changes may be clinically relevant.

To test whether PD is associated with systemic mitochondrial dysfunction we used ^31^P MRS to measure in vivo ATPmax in hand and leg skeletal muscles. Several studies have demonstrated that in vivo ATPmax is associated with poor physical function with age, although there is a lot of variation in mitochondrial function in older adults [18,19,41]. To test whether PD is associated with skeletal muscle mitochondrial dysfunction independent of age, we compared in vivo ATPmax in hand and leg from 30 newly recruited PD subjects with a dataset of healthy control subjects collected as part of the screening process for previous studies at the Translational Bioenergetics Laboratory. In vivo ATPmax provides a direct estimate of the capacity for mitochondrial ATP production under physiological conditions in the skeletal muscle. The significantly lower ATPmax in the TA in the PD subjects indicates that, although the primary functional defects in PD are believed to originate from the CNS, the mitochondrial dysfunction extends to peripheral tissues. The absence of a difference between healthy controls and PD subjects in ATPmax in the hand indicates that the extent of mitochondrial dysfunction, while not confined to the CNS, appears to vary among tissues.

Although ATPmax provides a measure of mitochondrial capacity, it does not provide much insight into the effect of PD on resting metabolism or metabolic homeostasis. To address this question, we attempted to measure nicotinamide levels from the ^31^P spectra. NAD metabolism has received a lot of attention due to its multiple roles in sirtuin-based deacetylation and PARP DNA damage and repair pathways [42]. In addition, NADH, the reduced form of NAD, is generated by the tricarboxylic acid cycle in the matrix of the mitochondria and provides the main electron input into complex I of the ETS to initiate oxidative phosphorylation. Disrupted NAD and NADH levels are associated with defective complex I function [43,44] and multiple chronic disease conditions, including Parkinson’s disease and Alzheimer’s disease models [45]. Most previous studies focused on nicotinamides have measured metabolite levels ex vivo in blood from human subjects or tissue biopsies in animal models. However, these metabolites are represented by small peaks on the shoulder of the alpha-ATP peak in the ^31^P spectra and thus can be assessed in vivo in brain and skeletal muscles [38,46]. In the current study, we were able to quantify NAD from spectra from the leg muscles only due to the lower signal quality coming from the smaller FDI muscles of the hand. Our data demonstrating significantly reduced NAD in PD subjects is consistent with a recent pilot study that observed altered in vivo NAD+ in the brains of early onset PD subjects [47]. Unfortunately, we were not able to quantify NADH values as reported for the brain for several reasons: (1) The NAD and NADH peaks are obscured by the much higher [ATP] in muscle (8.2 mM) relative to brain (2.8 mM) [48]; (2) ATP and nicotinamides in muscle appear to have longer correlation times/shorter T_2_’s than brain. For example, Lu et al. reported just under 8 Hz line width average at 4 T for alpha-ATP [49].

Report values ranging from ~3 to ~9, whereas Ren et al. 4–8 [49,50], while values for the NAD/NADH in skeletal muscle are much higher, with estimates as high as 540 for cytosol or six for mitochondria [51].

Despite the inability to quantify the NAD redox ratio, data showing the decrease in NAD in the skeletal muscle of PD subjects is an important new insight because of the central role in both metabolism and stress response pathways. Interventions to manipulate NAD levels based on supplementing steps in NAD synthesis or salvage in the cell are now a major industry (e.g., NR, NMN, NAM). Elevating NAD in animal models and cells from idiopathic PD patients supports investigating NAD as a therapeutic target. In cultured neurons, NAD shows effects of axonal protection [52]. Nicotinamide riboside (NR), a precursor of NAD^+^, prevents noise-induced hearing loss and degeneration of neurites in the spinal ganglion [53]. In patient-derived induced pluripotent stem cells, increasing NAD^+^ with NR significantly ameliorated mitochondrial dysfunction in neurons [54]. In fly models of *GBA*-related PD, NR prevented loss of dopaminergic neurons and motor decline. The authors conclude that NR is a viable clinical avenue for PD neuroprotection research [29]. The biology of NAD in PD and results from animal and cell models has led to multiple ongoing clinical trials for NAD targeted interventions in PD [28]. To fully develop NAD targeted strategies, it will be important to be able to measure changes in nicotinamide levels in vivo as well as to assess the effects of these changes on PD pathology. Results reported here suggest that ^31^P MRS in skeletal muscle may be an important tool in the development and testing of NAD based interventions.

Muscle fatigue and weakness are two of the most common and debilitating symptoms of PD. The underlying factors driving muscle fatigue and weakness in PD are not well understood. However, most focus has been on the role of central factors as a primary cause, with some suggestions of a role for circulating factors. In the leg muscles in this study, we have identified significant muscle weakness and reduced fatigue resistance in the skeletal muscles of PD patients associated with low ATPmax and disrupted metabolic homeostasis. Although this study was not designed to test mechanisms of muscle dysfunction, the metabolic defects in the muscles suggest that there is a potential role for skeletal muscle intrinsic pathology in PD-related muscle dysfunction. However, the absence of a decline in ATPmax in the hand despite similar weakness and fatigue to that observed in the leg supports an important role for central factors as drivers of muscle fatigue in PD.

The higher the leg NAD, the less RBD and apathy reported by participants. Both hand and leg ATPmax levels were associated with the motor subset (part 3) of the UPDRS, and hand ATPmax was associated with mild, but statistically significant improvement in total UPDRS scores. Greater leg ATPmax was associated with less patient-reported symptom severity, specifically slowness, constipation, gait, falls, tremor, balance, visual disturbances, and nausea. This was an exploratory analysis and therefore no adjustments were made for multiple comparisons. While some of these associations may be due to chance, the consistent association between higher ATPmax and fewer motor symptoms, as measured by both the UPDRS part 3 and the motor subset of the PRO-PD, is notable.

This study has several limitations that should be considered when evaluating these results. In this study, it was not possible to specify whether the metabolic dysfunction we observed was a primary effect of PD or a secondary effect due to behavioral and mobility changes caused by PD. All of the PD subjects in this pilot study were mid-stage with Hoehn and Yahr scores of 2–3, indicating that all the subjects were experiencing PD symptoms and motor deficits. As an initial attempt to address this, we performed a sub-group analysis to compare results for ATPmax and muscle fatigue between subjects with either left- or right-side onset of symptoms. All of the measurements were made on the right side. Neither ATPmax nor muscle fatigue showed any effect of the side of onset in this analysis.

This study enrolled individuals with mid-stage PD; future studies should determine whether these findings are seen in the prodromal phase of PD and immediately following diagnosis or whether individuals in late-stage disease have even greater reductions in mitochondrial function. Skeletal muscle metabolism and performance are closely tied to activity level and nutritional status, which this pilot study was not powered to evaluate. Future studies should attempt to evaluate whether exercise frequency and intensity or nutritional status are related to mitochondrial function. Subsequent studies should also aim to determine whether these findings are unique to idiopathic PD or observed in other forms of parkinsonism as well. Finally, this pilot study compared newly recruited PD subjects with the historical control dataset that we had collected at the UW; future studies should enroll healthy controls as a quality control measure, as reference ranges have not been established.

Following the demonstration of dopamine deficiency in PD, dopamine augmentation strategies have become an essential part of PD management. Similarly, the recent identification of metabolic deficiencies of NAD and ATP in the PD brain [47] and now skeletal muscle in vivo makes a compelling case for mitochondrial augmentation as a therapeutic strategy. MRS may have utility as a theragnostic biomarker in drug development, where it may be able to detect and monitor an intervention’s ability to improve mitochondrial function, as was recently demonstrated in the NADPARK study [30]. These are the first data to demonstrate an association between mitochondrial impairment of function and muscle function and worse PD symptoms, specifically motor symptoms.

## Figures and Tables

**Figure 1 nutrients-15-00943-f001:**
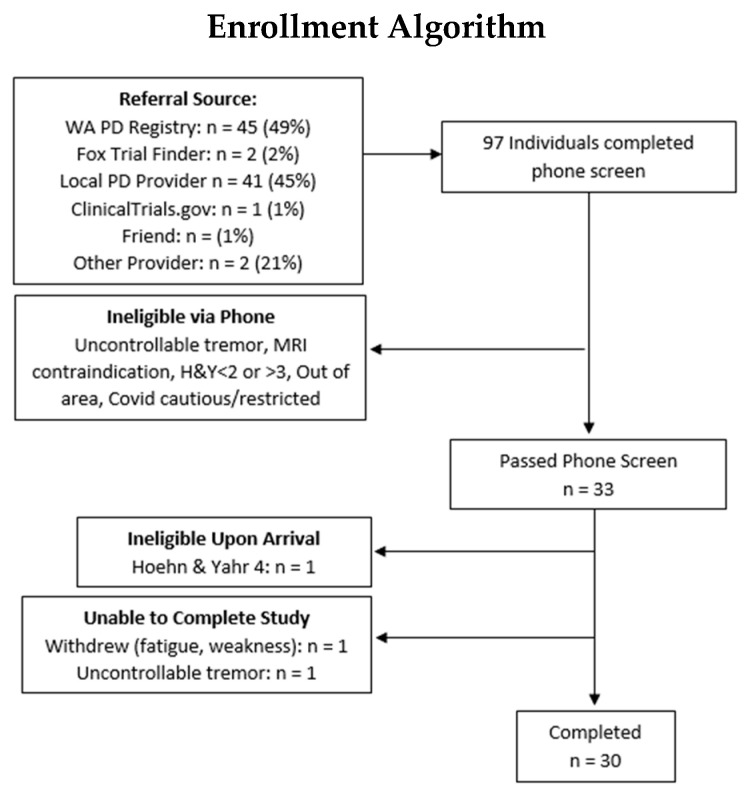
STROBE Flow Diagram of Experimental Model and Subject Details. 97 individuals were screened for inclusion over the phone, 64 of whom were determined to not meet study inclusion criteria. Of the 33 individuals invited to attend the in-person screening, one was disqualified upon arrival and two were unable to complete the study. See Appendix A for additional information on inclusion and exclusion criteria.

**Figure 2 nutrients-15-00943-f002:**
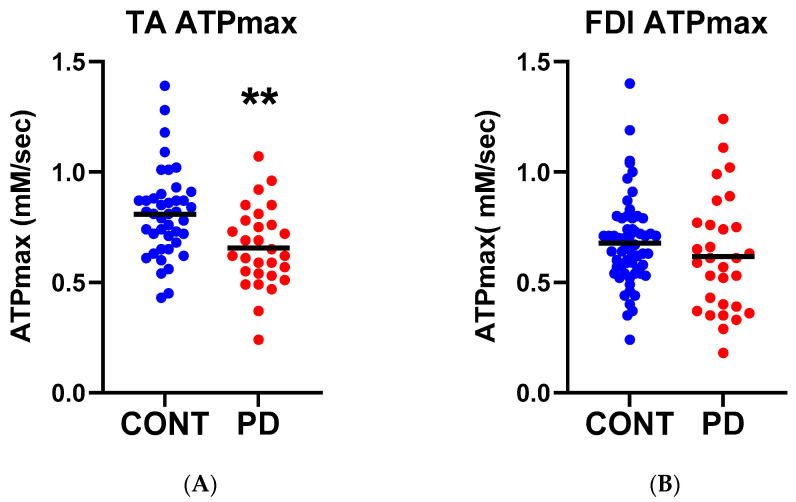
In vivo mitochondrial ATP production. Muscle maximum ATP production (ATPmax) measured using ^31^P MRS in the (**A**) leg (tibialis anterior, TA) and (**B**) hand (first dorsal interosseus, FDI). Solid line indicates mean for each group. ** *p* < 0.01.

**Figure 3 nutrients-15-00943-f003:**
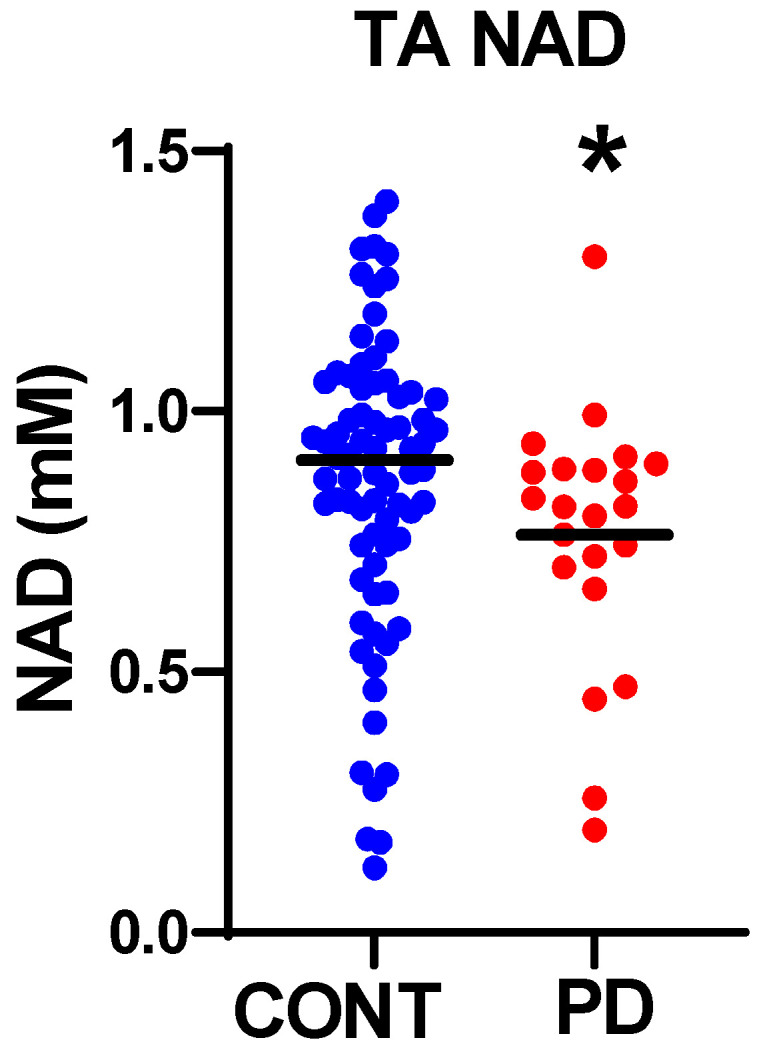
In vivo NAD in skeletal muscles. In vivo NAD concentration measured from ^31^P MRS spectra. Solid line indicates mean of each group. * *p* < 0.05.

**Figure 4 nutrients-15-00943-f004:**
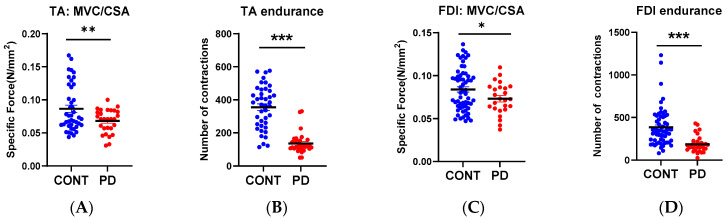
Muscle function in leg (TA) and hand (FDI). Comparison of muscle function between control and subjects with Parkinson’s disease. Muscle specific force in the (**A**) TA and (**C**) FDI; Muscle endurance measured as number of voluntary muscle contractions in the (**B**) TA and (**D**) FDI. * *p* < 0.05, ** *p* < 0.01, *** *p* < 0.0001.

**Table 1 nutrients-15-00943-t001:** Sample descriptive statistics.

	N = 33 (100%)
Characteristics	
Male, n (%)	24 (73%)
Female, n (%)	9 (27%)
Age, mean	72.5
Symptom Severity	
Years since PD diagnosis, mean	7.56
Hoehn & Yahr 2, n (%)	6 (22%)
Hoehn & Yahr 2.5, n (%)	11 (40.7%)
Hoehn & Yahr 3, n (%)	10 (37%)
UPDRS-Part 1, mean ± SD	2.85 ± 1.94
UPDRS-Part 2, mean ± SD	10.93 ± 4.98
UPDRS-Part 3, mean ± SD	16.74 ± 7.07
UPDRS-Part 4, mean ± SD	4.04 ± 2.69
UPDRS total, mean ± SD	34.59 ± 12.62
PRO-PD	
Mean (SD)	815 (423)
Minimum	275
Maximum	1763
Quality of Life	
Excellent	5 (18.5%)
Very Good	10 (37%)
Good	9 (33.3%)
Fair	3 (11.1%)
Poor	0

**Table 2 nutrients-15-00943-t002:** Muscle size, force production and bioenergetic properties for the tibialis anterior (TA) from control and PD subjects.

	PD (Mean ± SE)	Control
Hand		
Muscle Size	259 ± 12.59 (n = 25) *	218 ± 6.8 (n = 60)
Muscle Force	18.81 ± 1.1 (n = 30)	19.79 ± 0.5 (n = 135)
ATPmax	0.61 ± 0.047 (n = 30)	0.71 ± 0.02 (n = 134)
Leg		
Muscle Size	1189 ± 44.9 (n = 28) **	1021 ± 19.07 (n = 124)
Muscle Force	76.99 ± 4.07 (n = 30) **	88.5 ± 1.9 (n = 145)
NADt [ATP = 8.2]	1.060 ± 0.056 (n = 22)	1.186 ± 0.0039 (n = 79)
NAD [ATP = 8.2]	0.766 ± 0.051 (n = 22) *	0.907 ± 0.038 (n = 79)
ATPmax	0.656 ± 0.03 (n = 29) **	0.76 ± 0.017(n = 112)

Statistical difference between patients with PD and controls via an unpaired t-test with correction for unequal variances. * *p* < 0.05. ** *p* < 0.01.

## Data Availability

Deidentified raw data for both the control groups and the Parkinson’s cohort for metabolic, muscle function, and Parkinson’s disease severity will be granted upon request for 5 years after publication. To request access to data please contact one of the corresponding authors with a description of the data requested, planned analyses, and institutional affiliation of the requesting parties.

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
