# Peer review of "ATP and NAD+ Deficiency in Parkinson’s Disease"

_nutrients, 2023, doi:10.3390/nu15040943_

Round 1
Reviewer 1 Report
I thank the opportunity to review the original manuscript entitled "ATP and NAD+ deficiency in Parkinson's disease", sent for publication in "Nutrients". This original manuscript describes an important pathophysiological role of mitochondrial dysfunction and NAD+ deficiency and energetic failure in idiopathic Parkinson's disease. Methods have been clearly presented and Discussion section is disclosed with high level for different readers, clinicians and researchers. Some points should be evaluated by the authors:
1. Have the authors performed similar analysis in other forms of parkinsonism, such as atypical presentations, or in other neurodegenerative disorders, such as Hereditary Spastic Paraplegia, Inherited Ataxias, or Inherited Neurometabolic Disorders? I think some of the possible results in these groups could aid to differentiate if the mitochondrial dysfunction observed is more likely to results from a common primary mechanism or as a result of secondary features.
2. Do the authors think that idiopathic PD patients in late stages of disease course or with more severe handicap could be associated with different results in their profiles?
3. I suggest authors to change the word "mutations" (line 326) to "variants".
4. Do the authors think that reduced ATP production associated with increased symptom severity results from a primary mitochondrial dysfunction or only as a secondary finding in the context of a multi systemic degenerative disorder?
Reviewer 2 Report
This is a well-done pilot proof-of-concept clinical study. The exploratory study results indicate the noble role of mitochondrial and metabolic dysfunction in Parkinson’s disease (PD). The study design is solid, with appropriate control participants. The results interpretation is adequate with proper language and claims. I am supportive of this paper however; a few aspects below will need to be clarified before it is considered for publication.
Introduction: Authors have mentioned that PD is associated with oxidative stress, inflammation, impaired autophagy, and mitochondrial dysfunction (3). Older adults above the age of 60 years are reported to have oxidative stress and mitochondrial dysfunction and supplementing older adults with GlyNAC corrects these defects (PMID: 35975308) should be discussed as how PD is different from older adults. All the study participants are between 65-85 years.
Multiple typos/errors need to be corrected:
Like citation is before the full stop. 2040.[1], Translational Sciences (ITHS) [34,35]. exercise apparatus [33,36].
Figure 1 quality needs to be improved. It is not clearly visible in the current form.
Check the typos and arrangements in table 1: Male, n (. %)
I am not able to access the_ Supplementary Materials: The following supporting information can be downloaded at: 447www.mdpi.com/xxx/s1, Figure S1: Comparing effect of side of onset of PD symptoms; Table S1: List 448 of inclusion and exclusion criteria for PD subjects; Table S2: PD symptom severity correlations with 449 Leg [NAD], Leg [ATP] and Hand [ATP] by simple linear regressions. Non-normally distributed 450 variables were square root transformed..
Please check if it is uploaded.
Study limitations should be separated by title and paragraph.
